# Resistance of Gram-Positive Bacteria to Current Antibacterial Agents and Overcoming Approaches

**DOI:** 10.3390/molecules25122888

**Published:** 2020-06-23

**Authors:** Buthaina Jubeh, Zeinab Breijyeh, Rafik Karaman

**Affiliations:** Pharmaceutical Sciences Department, Faculty of Pharmacy, Al-Quds University, Jerusalem P.O. Box 20002, Palestine; bjubeh@gmail.com (B.J.); z88breijyeh@gmail.com (Z.B.)

**Keywords:** resistance, Gram-positive, MRSA, β-lactam, antimicrobial, antibiotic, bacteriophage, probiotic

## Abstract

The discovery of antibiotics has created a turning point in medical interventions to pathogenic infections, but unfortunately, each discovery was consistently followed by the emergence of resistance. The rise of multidrug-resistant bacteria has generated a great challenge to treat infections caused by bacteria with the available antibiotics. Today, research is active in finding new treatments for multidrug-resistant pathogens. In a step to guide the efforts, the WHO has published a list of the most dangerous bacteria that are resistant to current treatments and requires the development of new antibiotics for combating the resistance. Among the list are various Gram-positive bacteria that are responsible for serious healthcare and community-associated infections. Methicillin-resistant *Staphylococcus aureus*, vancomycin-resistant *Enterococcus faecium*, and drug-resistant *Streptococcus pneumoniae* are of particular concern. The resistance of bacteria is an evolving phenomenon that arises from genetic mutations and/or acquired genomes. Thus, antimicrobial resistance demands continuous efforts to create strategies to combat this problem and optimize the use of antibiotics. This article aims to provide a review of the most critical resistant Gram-positive bacterial pathogens, their mechanisms of resistance, and the new treatments and approaches reported to circumvent this problem.

## 1. Introduction

Alongside the discovery of antibiotics, resistance was always acknowledged and continuously developed. Sulfonamides were discovered in 1937 and resistance was reported in the late 1930s. Moreover, penicillin was discovered in 1928, and a few years later bacterial penicillinase was identified. Penicillin resistance led to the discovery of more β-lactams, but unfortunately, each discovery was consistently followed by the emergence of resistance. Increasing resistance to antibiotics led to a decline in the treatment options available to patients and consequently resulted in increased morbidity and mortality [1,2].

The World Health Organization (WHO) published a global priority pathogens list and categorized them as critical, high, and medium antibiotic-resistant bacteria that urgently need research and development of new treatments. Among these pathogens, Gram-positive bacteria which can cause serious infections and are considered a major concern and a health care problem, especially multidrug-resistant (MDR) bacteria like methicillin-resistant *Staphylococcus aureus* (MRSA), vancomycin-resistant *Enterococcus faecium* (VRE), and β-lactamase-resistant *Streptococcus pneumonia* [3,4].

Gram-positive bacteria can be identified using crystal violet dye which interacts with the bacteria to yield blue color under a microscope examination; this refers to the ability of the thick peptidoglycan (PG) layer to retain the dye [5]. The cell wall of Gram-positive bacteria (illustrated in Figure 1) differs from that of Gram-negative in which it lacks the outer membrane and have a thick layer of PG that surrounds the plasma membrane to protect Gram-positive bacteria from the harsh environment where they live in [6]. PG synthesis, thickness, chemical composition, and the extent of cross-linking can determine the cell shape of the bacteria and their morphology. Bacteria can be a sphere in shape such as staphylococci and streptococci, or rod-shaped like *Bacillus subtilis*, in addition to branching filaments bacteria [7]. Furthermore, the cell wall consists of long anionic polymers called teichoic acids, membrane protein which serves as sensors and a passage that facilitate the movement of different molecules, and capsular polysaccharides that are covalently attached to PG [8].

## 2. Resistance of Gram-Positive Bacteria

The resistance mechanism of Gram-positive bacteria can occur through two major strategies: enzymatic degradation of antibiotic by the production of β-lactamases, or by decreasing the affinity and susceptibility of their target site, the penicillin-binding protein (PBP), by either acquisition of exogenous DNA or by changes in the native PBP genes [9,10]. Figure 2 showcases the mechanisms of resistance of Gram-positive bacteria as related to the site of occurrence within the bacterial cell.

Several classes of antibiotics have a different mechanism of action. Examples for such mechanisms include (1) β-lactams target PBP and catalyze the last step of cell wall synthesis which leads to cell death. The resistance to bacteria results from the inactivation of antibiotics by β-lactamases. Penicillin resistance occurs mainly by the horizontal spreading of penicillinase plasmids by bacteriophages or by horizontal gene transfer that involves the PBP genes. On the other hand, methicillin resistance can result from additional PBP like PBP2/2a that is acquired from foreign DNA elements [9,11,12]. (2) Glycopeptides, like vancomycin and teicoplanin, act on inhibiting the last stage of cell wall synthesis. Resistance occurs by the acquisition of *van* gene cluster (*VanA*, *VanB*, *VanC*, *VanD*, *VanE*, and *VanG*) which results in low binding affinity to glycopeptides [10,13]. (3) Quinolones act by inhibiting DNA gyrase and topoisomerase IV which causes death to bacteria. Mutation in the subunits of these two enzymes (grIA/grIB and gyrA/gyrB) results in quinolone resistance [14,15]. (4) Aminoglycosides bind on the 30S ribosome which inhibits translocation and results in nonfunctional proteins that disturb the membrane structure and increase aminoglycoside penetration. Resistance can occur by the acquisition of aminoglycoside modifying enzymes, such as nucleotidyltransferases, acetyltransferases, and phophotransferases, or by mutations and efflux mechanism [16]. (5) Macrolides bind on the 50S ribosome and inhibit protein synthesis. Resistance developed by different mechanisms such as methylation of 23S rRNA, efflux systems (Mef(A), Msr(A)), and mutation in 23S rRNA and protein L4 [10,17]. (6) Oxazolidinones like linezolid inhibit protein synthesis by binding on the 50S ribosome subunit. Resistance is mainly due to mutation in 23S rRNA and G2576T in DNA [9,10,12].

In this review, the resistance of Gram-positive pathogens is discussed according to their categories by the WHO.

### 2.1. Staphylococcus Aureus 

*Staphylococcus* is a genus of the Gram-positive cocci family *Staphylococcaceae*. S. aureus is a major human pathogen associated with high infection and mortality rates and is one of the leading causes of minor and life-threatening diseases like infections of the respiratory tract, skin and soft tissue, pleuro-pulmonary, device-related infections, and infective endocarditis [18,19].

Good susceptibility to *S. aureus* led to the discovery of penicillin by Alexander Fleming. Penicillin resistance was developed just a few years after its introduction into the clinical practice and within a decade it became a huge problem in the community. *S. aureus* has an extraordinary ability to acquire resistance to any antibiotic. Penicillin-resistant *S. aureus* produced a plasmid-encoded penicillinase that hydrolyzes the β-lactam ring of penicillin which is essential for its antimicrobial activity [20].

#### 2.1.1. *Staphylococcus Aureus*—Methicillin-Resistant

In 1959, celbenin or currently called methicillin semisynthetic penicillin was developed to counteract a bacterial resistance mechanism, but in 1961 the first methicillin-resistant *S. aureus* strain was identified in the UK and was found to be resistant to all β-lactam antibiotics including cephalosporins and carbapenems. The resistance is due to the production of an additional PBP, designated PBP2a, with a reduction to the affinity for penicillin and β-lactam antibiotics. PBP2a is the product of the genomic*A* which is acquired by *S. aureus* from unknown heterologous sources [18,21,22].

Methicillin-resistant *Staphylococcus aureus* (MRSA) outbreaks led to its classification into two types: (1) healthcare-associated MRSA (HA-MRSA) in which patients exposes to MRSA strains due to hospitalization, surgery, hemodialysis, etc., and is typically resistant to clindamycin, and (2) community-associated MRSA (CA-MRSA) is described as strains that can infect healthy people that have no contact with healthcare facilities, and they are susceptible to clindamycin and none β-lactam antibiotics [23].

#### 2.1.2. *Staphylococcus Aureus*—Vancomycin Intermediate and Resistant

MRSA infections burden led to intensive use of the antibiotic vancomycin which is a glycopeptide antibiotic caused for the appearance of vancomycin-intermediate *S. aureus* (VISA) which is not inhibited *in vitro* at vancomycin concentration below 4–8 μg/mL. Vancomycin-resistant *S. aureus* (VRSA) is inhibited only at concentrations of 16 μg/mL or more [20]. VISA and VRSA have emerged from MRSA; however, VRSA does not progress from VISA because both have different resistance mechanisms. VISA was first observed in Japan in 1996 and was associated with the presence of a thickened cell wall which is rich in peptidoglycan chains that are not cross-linked and offer the terminal dipeptide D-Ala-D-Ala which is the target of vancomycin. Vancomycin’s effective target is the D-Ala-D-Ala residues belong to the peptidoglycan precursors in the cell membrane [24,25]. Therefore, the D-Ala-D-Ala residues in the thickened cell wall act as decoy targets that block vancomycin in the external layer of the cell wall and divert it from reaching its true targets. On the other hand, VRSA acquired the complete genetic resistance from vancomycin-resistant *enterococci* (VRE). Six VRSA strains have been identified in the USA and all have acquired the *vanA* genes that give a high resistance to glycopeptides, vancomycin, and teicoplanin. *VanA* genes encode the synthesis of modified peptidoglycan precursors containing a terminal D-Ala-D-Lac in which vancomycin has much lower affinity compared to the terminal wild type D-Ala-D-Ala [22].

#### 2.1.3. *Staphylococcus Aureus*—other Antibiotic Resistance

In the 1990s, after the introduction of ciprofloxacin, resistance to fluoroquinolones has emerged rapidly in *S. aureus*, especially in MRSA. Resistance developed due to mutation in the genes encoding target enzymes that are essential for DNA replication (mutation in subunit gyrB of DNA gyrase and subunit grIA of Topoisomerase IV) and due to changes in drug entry and overexpression of an efflux pump NorA [26].

Linezolid and daptomycin were introduced as new anti-staphylococcal antibiotics licensed to treat MRSA infections. Linezolid is an oxazolidinone antibiotic which inhibits protein synthesis by binding to domain V of the 23S subunit of the bacterial ribosome. The drug was approved in 2000 for nosocomial infections caused by MRSA [27]. Resistance to linezolid was found to be as a result of single-nucleotide mutation in the binding site for linezolid. Daptomycin is a cyclic peptide antibiotic with fatty acid side chain which was licensed for the treatment of *S. aureus* bacteremia and endocarditis. It’s mechanism of action involves binding and insertion into the bacterial cytoplasmic membrane in the presence of calcium ions. Daptomycin resistance was developed due to a mutation in at least three distinct proteins. The mechanism of this resistance involves an increased voltage difference across the cytoplasmic membrane and reduced drug binding to its target site [28,29]. All emergencies of VISA and VRSA in hospitals and community-acquired MRSA have led to the use of second-line anti-staphylococcal antibiotics (trimethoprim-sulphamethoxazole and tetracyclines) as alternative or adjuvant in combination with β-lactams or vancomycin. Despite the susceptibility of *S. aureus* to these drugs, acquired resistance can happen. Sulfonamide inhibits dihydropteroate synthase (DHPS), which condenses pteroate and p-aminobenzoic acid (PABA) to form dihydropteroate, a precursor of folic acid. The resistance to sulfamethoxazole is due to chromosomal encoded DHPS mutation which prevents the drug from binding to the enzyme. Trimethoprim target is dihydrofolate reductase (DHFR). It is used clinically in combination with sulfamethoxazole and neither of them is used in monotherapy. Resistance to trimethoprim is due to the acquisition of the *dfrA* gene that encodes DHFR enzymes that are not susceptible to inhibition [27,30,31]. Tetracyclines inhibit protein synthesis by binding to the 30S subunit. Resistance to tetracycline happens by two different mechanisms: firstly, by active efflux of the drug which encoded by plasmid born genes *tet*(K) and *tet*(L), and secondly by ribosome protection, the tetracycline target site encoded by the genes *tet*O/M [32]. Fusidic acid is a topical preparation used to treat *S. aureus* skin infections. Resistance to this antibiotic emerges rapidly by a mutation in the chromosomally located *fusA* gene and also by acquisition of the *fusB* determinant. Fusidic acid and rifampicin monotherapy caused rapid single mutation, that is why they are used together as a combination to treat MRSA infection and as an alternative to linezolid [22,27].

Clindamycin antibiotic is used against MRSA, especially CA-MRSA. Resistance to this drug rises from genes designated *erm*, which encodes methylation of an adenine residue in the 23S subunit of the bacterial ribosome. This leads to resistance to macrolide, lincosamide, and streptogramin antibiotics, the MLSB resistance [22,33]. Aminoglycosides are used to treat staphylococcal infections like endocarditis, in combination with antistaphylococcal penicillin or vancomycin. Resistance developed due to enzymatic modification of aminoglycosides and as a result, it cannot bind to the ribosomal target site and cannot block protein synthesis [22].

### 2.2. Enterococcus Faecium

Enterococci are Gram-positive cocci. There are many Enterococcus species but only two are responsible for human infections including *Enterococcus *faecalis** that is responsible for 80 to 90% of all clinical isolates, and *E. faecium* responsible for 5 to 15% *E. faecium* is part of the normal flora in human and animal guts, but in immune-compromised hosts, *E. faecium* can act as an opportunistic pathogen which can cause severe morbidity and mortality [34,35].

#### 2.2.1. *E. faecium*—Ampicillin/Penicillin and Cephalosporins resistance

Resistance of *E. faecium* to β-lactams like penicillin was found to be associated with the presence of *pbp5* chromosomal gene which encodes a low binding affinity class B PBP for ampicillin/penicillin and the cephalosporins. In addition, mutated PBP and the overexpression of β-lactamase enzymes results in high resistance levels to β-lactam antibiotics. Other pathways are also reported to be involved with cephalosporins resistance such as cognate response regulator CroR, serine/threonine kinase designated IreK, and phosphatase IreP [36,37].

#### 2.2.2. E. faecium—Vancomycin-Resistant

In the 1970s, enterococci developed an intrinsic resistant to third-generation cephalosporins and later to ampicillin which led to the introduction of vancomycin as a treatment option; however, vancomycin-resistant enterococci (VRE) have emerged and in 1990s become the second most common nosocomial pathogen due to heavy use of vancomycin. *E. faecium* can acquire genes through mobile genetic elements such as plasmids which were categorized into 6 of the 19*rep*families and transposons such as*Tn1547*which give vancomycin type B resistance [35].

Vancomycin acts by binding to the D-alanyl-D-alanine (D-Ala-D-Ala) terminus and inhibits cell wall synthesis. Vancomycin-resistance gene clusters (such as, van A, B, D, and M) are responsible for the replacement of D-Ala-D-Ala with D-alanyl-D-lactate termini which results in low binding affinity of vancomycin. Van A gene cluster is the most common type, and was found on transposon that is related to*Tn1546* [38].

#### 2.2.3. E. faecium—other Antibiotic Resistance

*E. faecium* is considered a MDR bacteria. The main resistant mechanisms of *E. faecium* to aminoglycoside like tobramycin, kanamycin, and gentamicin involve aminoglycoside-modifying enzymes (AMEs) including aminoglycoside nucleotidyltransferases (ANTs) aminoglycoside acetyltransferases (AACs) and aminoglycoside phosphotransferases (APHs). Moreover, acquisition of genes encoding ANT(3″)-Ia or ANT(6´)-Ia enzymes and single-step mutations in the S12 ribosomal protein can result in high level resistance to streptomycin. High resistance levels of *E. faecium* to fluoroquinolones are also reported; point mutations in gyrA and parC genes that encode subunits A of DNA gyrase and topoisomerase IV or NorA-like efflux pump results in high resistance levels to FQs. Furthermore, resistance to Quinupristin–dalfopristin, a streptogramin drug that interfere with bacterial protein synthesis (23-S rRNA of the 50S ribosomal subunit), can be developed through drug efflux, modifying enzymes, or modification of the ribosomal target [39].

### 2.3. Streptococcus Pneumoniae

*Streptococcus pneumoniae* is a Gram-positive bacteria and one of the most important pathogens that has the ability to colonize the upper respiratory tract and cause infections such as meningitis, sinusitis, bronchitis, pneumonia, and others.

#### 2.3.1. Streptococcus Pneumoniae- Penicillin-non-Susceptible

Pneumococcal infections are increasing rapidly due to the development of antimicrobial resistance which started to appear in 1912 [40,41]. However, it was not until 1965 that penicillin resistance *S. pneumoniae* (PRSP) was reported. Resistant to three or more classes of antimicrobials was referred to as MDR pneumococci [42,43]. Penicillin resistance occurs due to alternations in one or more of the six PBPs found in the *S. pneumoniae* cell membrane. This may due to chromosomal mutation or acquired by natural transformation in which a genome is picked up from other bacteria and incorporated into pneumococci DNA. Children, elderly, and daycare center’s attendance are at high risk of being colonized and infected with resistant pneumococci [44,45].

#### 2.3.2. *S. Pneumoniae*-other Antibiotic Resistance

In addition to penicillin resistance, pneumococci resistant to erythromycin and trimethoprim-sulfamethoxazole (TMP-SMX) are widely spread. Macrolide–lincosamide–streptogramin B resistance may be mediated by *erm(B)* gene that encodes a methylase, or by *mef(A)* gene that encode antibiotic efflux pump. Other resistance includes ribosomal RNA (23S rRNA), ribosomal proteins L4 and L22 mutations. Other resistance was also reported to tetracycline, chloramphenicol and low levels of fluoroquinolone resistance [43,46].

### 2.4. Other Resistant Gram-Positive Bacteria

▪*Staphylococcus epidermidis* uses exopolysaccharide matrix or its ability to form biofilms as a mechanism of resistance to reduce penetration and permeability of antibiotics. The majority of *S. epidermidis* isolates from different nosocomial infections are methicillin-resistant strains due to the transfer of resistant *mecA* gene that encode PBP2a, in addition, resistance to quinolones and vancomycin were also reported [47].▪*Staphylococcus saprophyticus* is the most common cause of uncomplicated urinary tract infections (UTIs). Resistance of *S. saprophyticus* occurred to commonly prescribed UTIs antibiotics such as ampicillin, ceftriaxone, cephalexin, and ciprofloxacin [48].▪*Streptococcus viridans* are an upper respiratory tract commensal bacterial that developed resistance to penicillin and other β-lactam antibiotics due to alteration in the penicillin-binding protein. In addition, other reports demonstrated that *Streptococcus viridans* can serve as reservoirs for resistance genes such as *mef*(E) and *mel* genes which develop resistant to the macrolide-lincosamide-streptogramin B (MLS(B)) antibiotics [49,50].▪*Streptococcus pyogenes*is a human pathogen that colonize in the upper respiratory tract and skin. *S. pyogenes* are resistant to macrolides, lincosamides, and streptogramins (MLS), in addition to tetracyclines, and very uncommon resistance to aminoglycosides or fluoroquinolones [51].▪*Streptococcus agalactiae* or Group B streptococcus (GBS) is responsible for most neonatal infections in humans that can be transferred from mother to child via the maternal genital tract into the amniotic fluid or at delivery. Resistance to antibiotic such as erythromycin and other macrolides is due to either ribosomal modification encoded by *erm* genes or through efflux pump mediated by *mefA* genes. Moreover, ribosomal translocation encoded by *linB* genes results in clindamycin resistance in GBS [52].▪Clostridium difficile and C. Perfringens

*Clostridium difficile* is the most common cause of healthcare antibiotic-associated diarrhea, and antibiotic resistance to *C. difficile* allows it to grow and colonize in the gastrointestinal tract. Resistance was found to be associated with hyper-virulent RT027 and RT078 strains that cause severe infections that require the massive use of antibiotics like fluoroquinolones (FQs), resulting in acquisition of resistance. Resistance of *C. difficile* strains to clindamycin and cephalosporins are also reported [53]. *Clostridium perfringens* is also a pathogenic bacterium that may come from contaminated food and causes gastrointestinal infections mediated by toxins release. Resistance of *C. perfringens* was developed to antibiotics that are used in farms such as streptomycin, lincomycin, trimethoprim-sulfamethoxazole, with lower resistance percentage to ciprofloxacin, cefotaxime, and rifampicin [54,55].

▪*Bacillus anthracis* and *Bacillus cereus* are spore-forming bacteria. *B. anthracis* causes anthrax disease by its virulence factors—capsule and toxin that are encoded on plasmids—while *B. cereus* is a soil bacterium and human pathogen that causes contamination to dairy industry by producing numerous enzymes and aggressins. *B. antracis* has a good susceptibility to penicillin in contrast to *B. cereus* which produce potent ß-lactamases and resistance against penicillin, ampicillin, cephalosporins, and trimethoprim [56,57].▪*Corynebacterium diphtheria* is a human pathogen that causes diphtheria disease, an upper respiratory tract illness mediated by potent A-B exotoxin named diphtheria toxin that inhibits protein biosynthesis and kills susceptible host cells. Horizontal gene transfer of antibiotic resistance genes such as *cmx*, *sul1*, and *tet*(W) result in resistant *C. diphtheria* to chloramphenicol, sulfonamides, and tetracyclines [58].▪*Listeria monocytogenes* is a foodborne pathogen that causes severe infections in humans that can be treated with early administration of aminopenicillin and gentamicin antibiotics. Resistance of *L. monocytogenes* to tetracyclines and fluoroquinolones developed due to acquisition of conjugative transposons and active efflux, respectively. A low level of resistant strains to streptomycin, chloramphenicol, macrolide, and trimethoprim has been reported [59].

## 3. New Treatments

Resistant Gram-positive bacteria for β-lactams, aminoglycoside, linezolid, and daptomycin drugs made the treatment of patients a very challenging mission. Despite this fact, most of these drugs are currently in use for the treatment of resistant Gram-positive bacteria [60].

New antibiotics have been approved in the last decade for the treatment of multidrug-resistant Gram-positive bacteria such as follows.

▪Cephalosporins: ceftaroline **(1)** and ceftobiprole **(2)** (Figure 3) which are 5th generation cephalosporins that inhibit cell wall synthesis by binding to PBP proteins with higher affinity than other β-lactam drugs. Ceftaroline is active against many Gram-positive organisms like MRSA, VRSA, *Streptococcus pyogenes*, and others, although resistance increase in MRSA sp. was reported. On the other hand, ceftobiprole is active against Gram-positive and Gram-negative microorganisms [60,61,62].▪Oxazolidinones: tedizolid phosphate **(3)** (Figure 3) is the first generation of oxazolidinones, acts by inhibiting protein synthesis by binding to the 23S rRNA on the 50S ribosomal subunit with greater potency and bioavailability than linezolid [63].▪Quinolones: besifloxacin (**4**), delafloxacin (**5**)**,** and ozenoxacin (**6**) (Figure 3) all act by inhibiting DNA synthesis by binding to DNA gyrase and topoisomerase IV. Besifloxacin is active against Gram-positive bacteria, especially *S. aureus*, *Staphylococcus epidermidis*, *S. pneumoniae*, and *Haemophilus influenzae*, and Gram-negative bacteria, while delafloxacin is active against *S. aureus*, *S. pneumoniae*, and fluoroquinolone-resistant strains except for enterococci. On the other hand, ozenoxacin is active against MRSA, MSSA, MRSE, and *S. pyogenes* and was approved by the FDA to treat impetigo caused by *S. aureus* and *S. pyogenes* [62].▪Omadacycline (**7**) (Figure 3) is a tetracycline analog that inhibits protein synthesis by binding on the 30S ribosomal subunit. It is active against a wide spectrum of bacteria such as resistant Gram-positive pathogens (MRSA, VRE, *S. pneumoniae*, *S. pyogenes*, and *Streptococcus agalactiae*), Gram-negative aerobes, anaerobes, and atypical bacteria [64,65].▪Glycopeptides: dalbavancin (**9**), telavancin (**10**) and oritavancin (**11**) are vancomycin (**8**) derivatives and analogs (Figure 4). Dalbavancin inhibits cell wall synthesis and has an additional lipophilic side chain that enhances its activity and potency against wide-spectrum of Gram-positive organisms such as MRSA, *S. pyogenes*, *Streptococcus anginosus*, and *E. faecalis* susceptible to vancomycin. Telavancin inhibits cell wall synthesis and is active against aerobic and anaerobic Gram-positive bacteria. Oritavancin acts by inhibiting cell membranes and also inhibits RNA synthesis. It is active against MSSA, MRSA, VRE, and VISA VRSA [60,62,66].

## 4. Approaches to Overcome Gram-Positive Resistance

When dealing with resistant and multidrug resistant pathogens, development of new antibiotics within old classes is not enough. The discovery and development of novel antibiotics that have new mechanisms of action is crucial, but as this process is full of challenges and almost certainty of the emergence of resistance to those novel antibiotics, exploration of additional approaches is highly necessary. The following describes the newest approaches and development to fight MDR Gram-positive bacteria.

### 4.1. Novel Antibiotics

#### 4.1.1. Teixobactin

A novel antibiotic named teixobactin (**12**) (Figure 5), produced by *Eleftheria terrae*, a species of β-proteobacteria, was discovered in 2015 in a screen of uncultured bacteria (bacteria that do not grow under laboratory conditions). The discovery of the drug was accomplished by screening the bactericidal activity of extracts of uncultured soil microorganisms after in situ cultivation in diffusion chambers. Teixobactin inhibits the synthesis of cell wall peptidoglycan by binding to highly conserved precursors like lipid II and lipid III and found to be very active and potent against Gram-positive bacteria including drug-resistant strains. Additional to its activity on Gram-positive bacteria, teixobactin showed a good activity against asmB1, a strain of *Escherichia coli* with a defective outer membrane permeability barrier [67]. No toxicity to mammalian cells was reported and no mutants of *S. aureus* or *Mycobacterium tuberculosis* resistant to teixobactin were obtained in laboratory resistance induction experiments, suggesting that developing resistance is expected to be difficult. In vivo studies in murine models indicated that teixobactin has the potential to be a good treatment for human MRSA infections [68].

#### 4.1.2. Malacidins

In 2018, Hover et al. reported the discovery of calcium-dependent antibiotics as a distinctive class of antibiotics called malacidins (**13**) (Figure 5), which are active against MDR Gram-positive bacteria [69]. Malacidins are natural products of soil microbiomes discovered by a culture-independent natural product discovery platform. This platform was developed by the Hover team in which they extracted DNA from soil samples and then performed sequencing, bioinformatic analysis and heterologous expression of biosynthetic gene clusters captured on the extracted DNA. They conducted a sequence-guided screen of more than 2000 soil collections for biosynthetic gene clusters encoding calcium-binding motifs. Malacidins inhibit bacterial wall biosynthesis by interacting with lipid II. Although calcium is essential for malacidins antibiosis, malacidins’ mechanism of action is distinct from that of previously known calcium-dependent antibiotics, daptomycin and friulimicin. Malacidins were potently active against Gram-positive pathogens even those are resistant to vancomycin. When tested in vivo, malacidin sterilized MRSA skin infection in rats after topical administration. Laboratory experiment to induce resistance has failed to detect resistance to malacidins. No significant toxicity against mammalian cells was observed at the highest concentrations tested [69].

#### 4.1.3. Antimicrobial Peptides

Antimicrobial peptides (AMPs) are an essential part of the innate immune response in humans and other higher organisms. These peptides contribute to the first line of defense against infections as they are targeted against prokaryotes. Antimicrobial peptides are found to have antimicrobial, anti-attachment and anti-biofilm properties, which makes them one of the agents that can treat chronic infections effectively [70,71]. As part of the innate immunity, AMPs exert direct microbial killing activity and an indirect effect by the mediation of the inflammatory response resulting in cytokine release, cell proliferation, angiogenesis, wound healing, and chemotaxis [71].

AMPs are a good candidate for a novel antibiotic class. They have favorable features like a broad spectrum of activity, low incidence of bacterial resistance, low toxicity for eukaryotic cells, a specific mode of action and fast killing kinetics [72]. Moreover, they have good stability in wide ranges of pH and temperature [73]. AMPs have consistently exhibited potent synergism with clinically used antibiotics such as vancomycin, azithromycin, polymyxin E, penicillin, ampicillin, β-lactams, doxycycline, daptomycin, teicoplanin, linezolid, ciprofloxacin, and clarithromycin [74].

AMPs exert their microbicidal activity by increasing permeation and causing cell lysis after targeting the cytoplasmic membrane. Most AMPs affect the transmembrane potential and result in cell death. There are many reported modes of action of permeation by AMPs that are based on structural properties like size, sequence, cationic nature, amphipathicity, and hydrophobicity. The most commonly cited model of membrane destabilization by AMPs is the carpet model, in which peptides partition into acidic and zwitterionic membranes and are then electrostatically attracted to the anionic phospholipid head groups at numerous sites; covering the surface of the membrane in a carpet-like manner [75]. The barrel-stave model, which involves the formation of membrane-spanning pores [76]; toroidal pore model; the detergent model; the sinking raft model; and the interfacial activity model are other models illustrated by experimental results [76,77,78]. Additional to membrane permeabilization, AMPs neutralize or disaggregate lipopolysaccharide, the main endotoxin responsible for Gram-negative infections. Therefore, AMPs collectively protect against sepsis. Resistance to AMPs is relatively rare due to their attraction to the negatively charged lipid bilayer structure of bacterial membranes [72].

As for AMPs’ mechanism of biofilm inhibition, the mechanism is yet to be reported. It is hypothesized that the synergism of AMPs and antibiotics disrupts the biofilm matrix and allows the AMPs to target bacterial cells in the biofilm. Other possible mechanisms of action of AMPs in inhibiting biofilm formation could be the interference of quorum sensing and inhibition of adhesion of bacterial cells on solid surfaces [73].

Venoms of insects and arachnids are a rich source of AMPs; many of them have been tested for their antimicrobial activity on bacteria and fungi [79]. The South American social wasp *Polybia paulista* has a venom with a large variety of AMPs, polybia-CP is one of them [80]. Polybia-CP (Pol-CP-NH2) is a 12-residue cationic amphipathic mastoparan-like AMP with the sequence: Ile-Leu-Gly-Thr-Ile-Leu-Gly-Leu-Leu-Lys-Ser-Leu-NH2. Polybia-CP has high activity against Gram-positive bacteria and poor activity against Gram-negative bacteria. It is toxic to human cells [81]. Like other AMPs, polybia-CP is membrane-active and it targets the bacterial membrane. Polybia-CP kills bacterial cells when it enters the bilayer of the membrane and takes a helical conformation, this enables the amino acid residue to interact with lipid molecules and disrupt the alignment of the lipids forming water channels that leads to cell death [82].

The low predicted helical content of polybia-CP along with the presence of hydrophilic serine residue next to the C terminus is believed to be the reason for the decreased activity of polybia-CP against Gram-negative bacteria. Torres et al. have reported a rational physicochemical feature-guided design of novel small cationic amphipathic AMPs with extensively enhanced potency to Gram-negative based on polybia CP. The designed AMPs were able to kill bacteria at nanomolar concentrations and when tested in vivo in mouse models, they exhibited anti-infective activity [83]. Agelaia-MPI, Polybia-MPII, and Polydim-I derived from the venom of wasps and Con10 and NBDP-5.8 derived from the venom of scorpions are all AMPs with antimicrobial, antiviral, and antibiofilm activity [79]. Agelaia-MPI was superior to the other AMPs in inhibiting MDR-*Acinetobacter baumannii* [84]. Most AMPs in clinical use or clinical studies are restricted to the topical application or intravenous administration because of their short half-lives, as they are susceptible to proteolytic degradation [85].

#### 4.1.4. Dodecyl Deoxy Glycosides (Antimicrobials that Target Membrane Lipid Polymorphism)

Bacterial cell membrane represents a good target for antimicrobial, due to its essentiality for cell viability regardless of the metabolic state of the cell, but the challenge is to find agents that are selective for microbial membranes. The fact that prokaryotic and eukaryotic membranes display some difference promoted the search for leads that target cell membranes with a high degree of selectivity toward prokaryotes. This led to the discovery of the family of dodecyl deoxy glycosides, which are sugar-based antimicrobials that target membrane lipid polymorphism [86].

Dodecyl deoxy glycosides interact with phosphatidylethanolamine (PE) of the membrane and induce membrane disruption through phospholipid lamellar-to-inverted hexagonal phase transition. The sugar-based surfactant dodecyl 2,6-dideoxy arabino-hexopyranoside has antibacterial activity against Bacillus species [87]. Based on this, Dais and coworkers have synthesized several deoxy glycosides and analogs and performed structure–activity relationship and mechanistic studies on them. Considering that bacterial membrane is rich in PE, unlike mammalian cells that are rich in phosphatidylcholine (PC), Dias et al. conducted biophysical studies on PE-rich vs. PC-rich membranes [86].

This research was directed to find new antibiotics for *Bacillus anthracis* which is resistant to the treatment of choice, ciprofloxacin, and forms a serious bioterrorism threat [88]. *Bacillus cereus* is a microbe genetically related to *B. anthracis* and was used instead of *B. anthracis* in the research due to safety issues [56]. It was found that the glycone of dodecyl 4,6-dideoxy-α-D-Xylo-hexopyranoside **(14)** (Figure 5) has conducted the highest bactericidal activity with low toxicity and good selectivity to prokaryotes. Deoxy C-glycosides of D-series with α-configuration showed activity but also exhibited toxicity [86].

#### 4.1.5. Cannabinoids

Cross-resistance to microbial and plant antibacterial agents are rare, which is why plants are still an important source of antimicrobial agents [89]. Cannabis sativa is an herbaceous species that have been used in folk medicine. It gained interest because of its multipurpose applications and its metabolites that showed potent bioactivities in human health. It has been known for a long time that cannabis saliva contains powerful antibacterial agents and was investigated since the 1950s as an active topical antiseptic for oral cavity and skin, but scientists could not define any specific constituent. However, pre-cannabidiol is a powerful antibiotic. Non-psychotropic cannabinoids and a psychotropic agent such as cannabichromene (**15**), cannabigerol (**16**), cannabidiol (**17**), cannabinol (**18**), and ∆9-tetrahydrocannabinol (**19**) (Figure 5) were found to have antibacterial activity. Appendino et al. investigated the antibacterial profile of these five alkylated and acylated cannabinoids and their carboxylic precursors “pre-cannabinoids” to obtain structure–activity data and to define the microbiocidal cannabinoid pharmacophore. The result showed that cannabinoids’ antimicrobial activity depends on the modifications of the terpenoid moiety and their relation with the n-pentyl chain which affects lipid affinity and cellular bioavailability antimicrobial agents [89]. Chakraborty et al. also examined the antimicrobial activity of *Cannabis sativa*, *Thuja orientalis*, and *Psidium guajava* present in the leaf extracts were found to inhibit MRSA growth. A synergistic effect was noticed when *C. sativa* was used in combination with *T. orientalis* and when *P. guajava* used with *T. Orientalis* in a 1:1 ratio for both. The inhibition is related to the combined inhibitory effect of phenolics and catechin found in the leaf extracts. Therefore, the result is promising in using these compounds to control hospital and community-acquired MRSA [90]. *C. Sativa* plant is an important source that contains high concentrations of non-psychotropic cannabinoids, and therefore they can be used against MDR in MRSA strains and other pathogenic bacteria which make them attractive antibacterial leads. More studies are needed to establish safety and environmental profile of cannabinoids and the ability to use them systemically [89].

#### 4.1.6. DCAP

In 2012, Hurley and coworkers introduced a new antimicrobial compound, DCAP (**20**) (Figure 5), discovered in a high-throughput screen of small molecules of in vitro inhibitory activity of the ATPase enzyme MipZ, the enzyme that regulates the placement of the division site in *Caulobacter crescentus*. DCAP(2-((3-(3,6-dichloro-9H-carbazol-9-yl)-2-hydroxypropyl)amino)-2 (hydroxymethyl)propane1,3-diol) is a potent antibiotic that kills a broad spectrum of Gram-positive and Gram-negative bacteria [91].

DCAP has two mechanisms of action that lead to bacterial membrane inhibition and cell lysis: it facilitates ion transport across the membrane, and consequently decreases the membrane potential. It also disrupts the lipid bilayer permeability [92,93]. DCAP is active against biofilms and dormant bacteria. It is a membrane-active agent distinct from other membrane-active agents in its specificity toward bacterial membranes. As for mammalian cells, DCAP does not affect the red blood cell membrane, and when tested for the effect on the viability of the mammalian cell, a decrease in the viability was observed only at high concentrations and after more than 6 h [92].

Two analogs of DCAP synthesized by Hurley et al. showed activity against *B. anthracis* and *Francisella tularensis* and synergistic antibiosis with ampicillin or kanamycin [93,94]. This indicates that DCAP and its analogs represent promising candidates for new antibiotic treatment for slow-growing and dormant bacteria.

#### 4.1.7. Odilorhabdins

Odilorhabdins (ODLs) **(21)** (Figure 5) are naturally produced peptides, produced by the enzymes of the non-ribosomal peptide synthetase gene cluster of *Xenorhabdus nematophila*, a nematode-symbiotic bacterium. ODLs represent a new class of antibiotics that is active against both Gram-positive and Gram-negative bacteria. ODLs are unique ribosome targeting bactericidal; these peptides inhibit protein synthesis by binding to the small subunit of bacterial ribosome at a site that is not exploited by existing antibiotics, increasing the affinity of non-cognate aminoacyl tRNAs to the ribosome, and inducing miscoding in the translation system. In vitro and in vivo studies showed promising efficacy and antibacterial spectrum, making ODLs an attractive starting point for clinical development [95,96].

### 4.2. Bacteriophage Therapy

In the early 1900s, Felix d’Herelle discovered types of viruses that have exclusive bactericidal activity, and named them bacteriophages [97]. Soon after, bacteriophages (or phages) were shown to be an effective treatment for bacterial infections [98,99], but the idea of using this type of treatment subsided by the surge of antibiotic discovery during antibiotic golden age. Recently, bacteriophage therapy is getting renewed interest after the crisis of conventional antibiotic resistance; there is an increasing interest in the discovery and development of new bacteriophages especially with the availability of genome sequencing and advanced modern technology [100].

Bacteriophages are self-amplifying, they kill bacteria by penetrating bacterial cells and disrupting many or all bacterial processes. At the same time, they are unable to penetrate eukaryotic cells, a fact that led to the safety of bacteriophages for human use. Bacteriophages are especially effective for the eradication of bacterial biofilms, they penetrate into biofilms by exploiting water channels within the biofilm to penetrate deep in the biofilm [101], or disrupt the extracellular biofilm matrix by expression of depolymerases [102], and they amplify while targeting dormant bacteria [103,104]. The only obstacle facing bacteriophage therapy development is the low number of studied patients and the lack of randomized control trials [103].

Bacteriophages augment the effect of antibiotics and have a high potential to be used as a combination therapy. Kuraman et al. studied the effect of bacteriophage (SATA-8505) and bacteriophage-antibiotic treatment against *S. aureus* biofilms. It was found that there was a significant reduction of viable biofilm-associated cells when the bacteriophage treatment preceded antibiotics, which provides a proof that bacteriophages have the ability to augment the activity of antibiotics against *S. aureus* biofilms [105]. Still further investigation and well-designed randomized clinical trials are highly needed to define the role of bacteriophages as a new treatment option, and to better understand the interaction between phages and antibiotics.

### 4.3. The Probiotic Approach to Prevent Antibiotic Resistance

Probiotics are defined as “live microorganisms which when administered in adequate amounts confer a health benefit on the host” [106], they mainly belong to the genera *Lactobacillus* and *Bifido bacterium*, among others [107]. Probiotics contribute to maintain the intestinal and other microbiota composition. Many probiotic strains have been documented to have beneficial effects in relation to antibiotic use; the main benefit is the reduction of the risk of antibiotic associated diarrhea. Concomitant use of probiotics with antibiotics reduces the incidence, duration and/or severity of antibiotic-associated diarrhea, which contributes to better adherence to the antibiotic prescription and consequently reduces the evolution of resistance [108,109]. Moreover, probiotics are documented to reduce the risk for infectious diseases in human and veterinary use [110], which potentially reduce the need for antibiotics

Probiotics are expected to contribute to reduce the spread and/or evolution of antibiotic resistance by assisting antibiotics, modulating immunity, reducing the need for antibiotics in some cases, and increasing antibiotic adherence. Probiotics assist antibiotics by the control of ability pathogenic bacteria, probiotics improve the intestinal barrier function [111], they also work by competitive exclusion by reducing pathogen’s coaggregation, adherence to cells, and antagonizing pathogenic bacteria by the production of organic acids [107]. Probiotics improve the function of antibiotics by producing antimicrobial compounds like bacteriocins, hydrogen peroxide, nitric oxide, and short chain fatty acids, thereby reducing the pathogenic bacterial populations and disrupting biofilms [112,113,114,115,116]. Additionally, probiotics can improve and stimulate the immune response, which in turn assist in the eradication of the pathogens at the mucosal site. Lactobacilli can affect macrophages proliferation and nitric oxide and cytokine production, especially interleukin (IL)-6, IL-10, IL-12 and tumor necrosis factor-alpha [117,118,119]. A strain of *Lactobacillus casei* from Argentina has been reported to increase phagocyte activity and secretory immunoglobulin A(IgA) [120].

The direct role of probiotics in preventing drug-resistant infections has not been established yet. So far, probiotics can be used as partial replacement or adjuvant to antibiotic treatment to help treating multidrug resistant infections [107]. It should be noted that probiotics must be specifically selected not to carry transferable antibiotic resistance and not to contribute to the spread of antibiotic resistance. For the future, more studies are needed to investigate the influence of probiotics on antibiotic use and antibiotic resistance.

### 4.4. National and International Approaches and Commitment

Antibiotic resistance is partly caused by human behavior, and the consequences affect everybody in the world. Global action is needed to face the danger of antibiotic resistance. Several strategies should be applied globally to combat bacterial resistance; some are already in practice in some countries and showed success in controlling resistance [121,122,123]. These strategies include surveillance of antibiotics to detect resistance in humans and animals, cautious use of antibiotics, decontamination or isolation of patients with resistant pathogens, improved antibiotic stewardship in healthcare facilities and community, restricted antibiotic advertising, good healthcare infrastructure, development of health insurance policies, development of diagnostic tools for prudent antibiotic prescription, and consistent disease control strategies [124,125]. One successful example on how national strategies affect resistance emergence is when the United Kingdom has implemented mandatory MRSA surveillance in 2001, the result was a significant reduction in MRSA bacteremia in hospitals in the UK [126]. Another good example is the efforts of the National Antimicrobial Resistance Monitoring System for Enteric Bacteria (NARMS) of the United States. which is a national public health surveillance system that tracks changes in the antimicrobial susceptibility of certain enteric bacteria found in ill people, retail meats, and food animals in the United States [125].

Unfortunately, in some regions, especially in low and middle income countries, there is a lag in the adoption of such strategies. Therefore, global coordinated actions need to be immediately taken in order to minimize the emergence of the problem and prevent facing medical, social, and economical setbacks [125]. Different global agencies and programs are putting efforts for the control of antibiotic resistance, like the Global Health Security Agenda (GHSA), Antimicrobial Resistance Action Package [127], the Global Antibiotic Resistance Partnership (GARP) [128], Food and Agriculture Organization, and the Global Antimicrobial Resistance Surveillance System under WHO [127].

### 4.5. Education of Prudent Antibiotic Use

One of the major drivers of antibiotic resistance is the inappropriate use and overuse of antibiotics consequent to insufficient knowledge. Education about prudent antibiotic use that targets both the prescribers and the public is of great importance. Most educational efforts target medical professionals, which have been influential in reducing antibiotic prescribing [129]. Interventions to improve the public’s antimicrobial resistance awareness and behaviors are proved to be effective in increasing the knowledge of antimicrobial resistance and engagement with antimicrobial stewardship behaviors. Despite this fact, the general public’s knowledge of antimicrobial resistance and their role in combating it is considered poor. Thus, effective engagement of the public (both adults and children) and cultural change is needed to motive the public to address antimicrobial resistance and participate in prudent antibiotic use [130]. This can be achieved by the development of educational programs suitable for each group, like educational lectures, campaigns, or patient information leaflets [129,130,131,132].

## 5. Conclusions

The resistance of pathogens toward current antibiotics is a spreading global crisis. Gram-positive bacteria include some of the most widespread resistant pathogens that form serious clinical challenges. MDR Gram-positive bacteria are major human pathogens, causing both community and healthcare-associated infections. Methicillin-resistant *Staphylococcus aureus*, vancomycin-resistant *Enterococcus faecium*, and drug-resistant *Streptococcus pneumoniae* are the major threats. Gram-positive pathogens exhibit an immense genetic ability to acquire and develop resistance to almost all clinically available antimicrobials. The main identified mechanisms of resistance include the alterations in the PBPs that lead to destruction of the active site of the antibiotic; prevention of the drug from reaching and binding to its target by modification of bacterial structures, like thickening of peptidoglycan and alterations of the ribosomal structure; or efflux of the drug by overexpression of efflux pumps. Constant, tremendous research to combat bacterial resistance nailed the discovery of innovative antibiotics and antibiotic classes with novel targets, such as teixobactin, malacidins, DCAP, and dodecyl deoxy glycosides. Moreover, referring to nature as a source of antimicrobials yielded some promising bactericidal candidates, like antimicrobial peptides, cannabinoids, and odilorhabdins. Bacteriophages and probiotics are two promising antibiotic alternatives that can also be used as a combinational treatment with antibiotics. In order to achieve effective resistance overcoming, combating strategies should be applied in community, national, and global levels. All these new strategies represent a hope for the future. Additionally, research efforts must be continued to investigate the constant evolving resistance, advance the development of the new candidates to reach clinical practice, and continue the search for new strategies to overcome resistance.

## Figures and Tables

**Figure 1 molecules-25-02888-f001:**
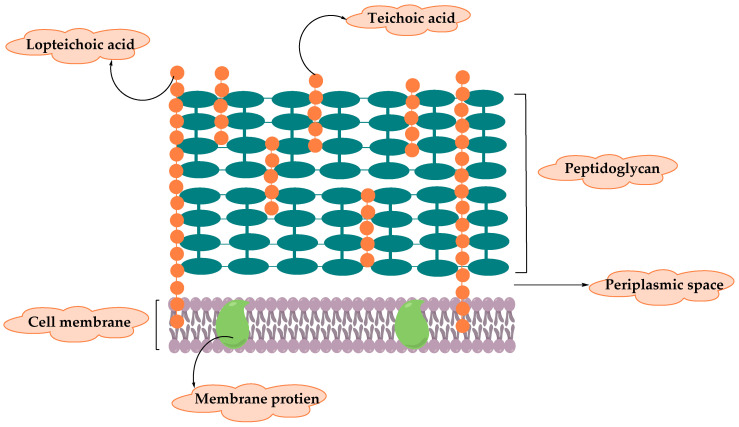
Diagram illustrating the cell wall structure of Gram-positive bacteria.

**Figure 2 molecules-25-02888-f002:**
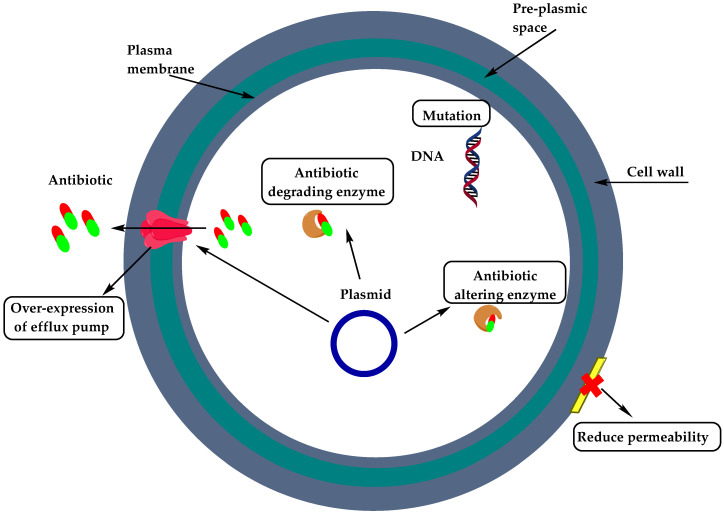
Mechanisms of resistance of Gram-positive bacteria.

**Figure 3 molecules-25-02888-f003:**
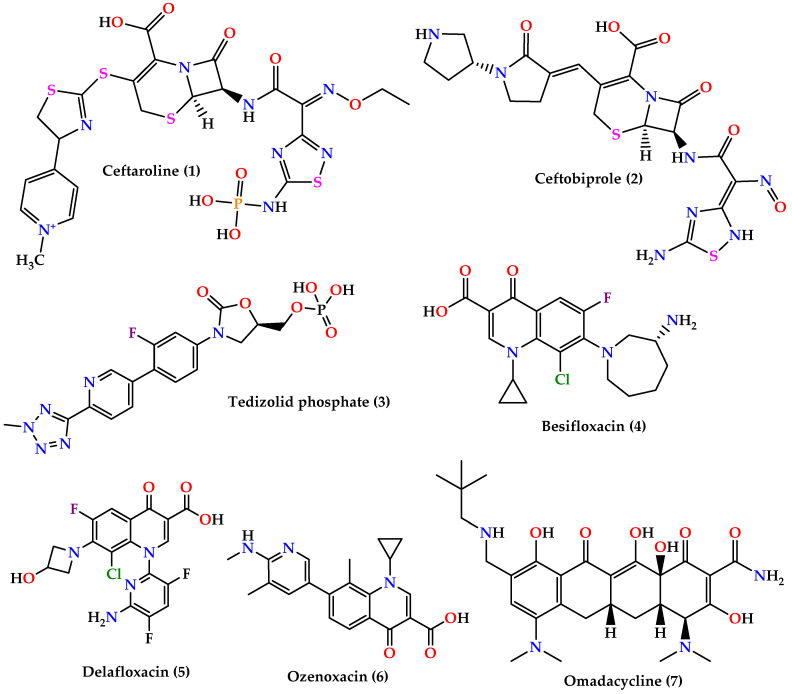
Chemical structures of ceftaroline (1), ceftobiprole (2), tedizolid phosphate (3),besifloxacin (4), delafloxacin (5),ozenoxacin (6), and omadacycline (7).

**Figure 4 molecules-25-02888-f004:**
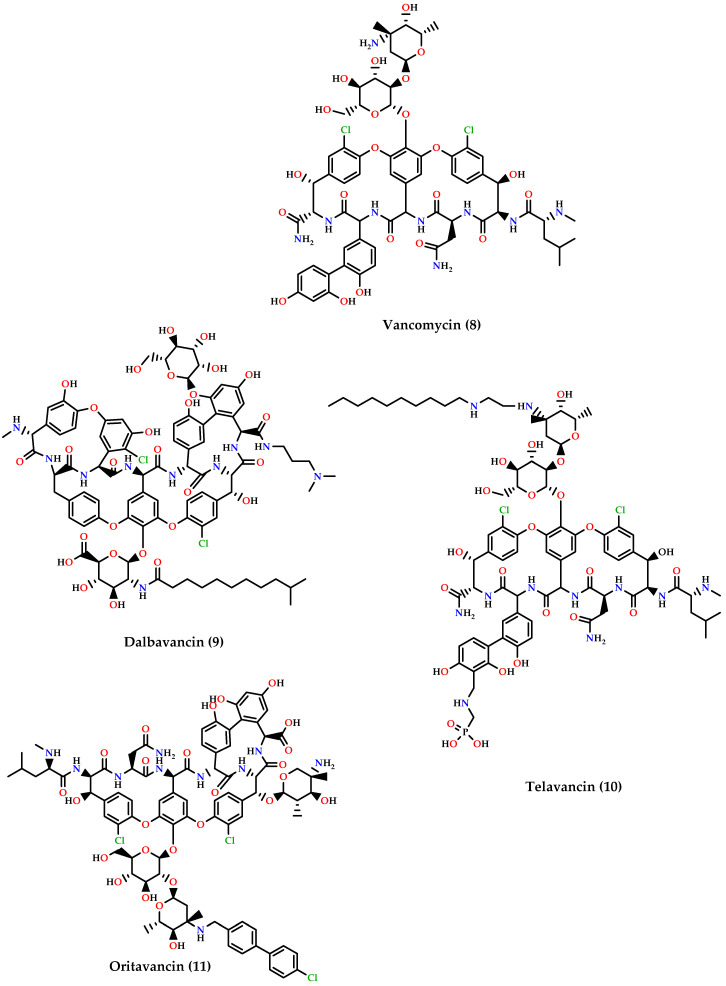
Chemical structures of vancomycin (8), dalbavancin (9), telavancin (10), and oritavancin (11).

**Figure 5 molecules-25-02888-f005:**
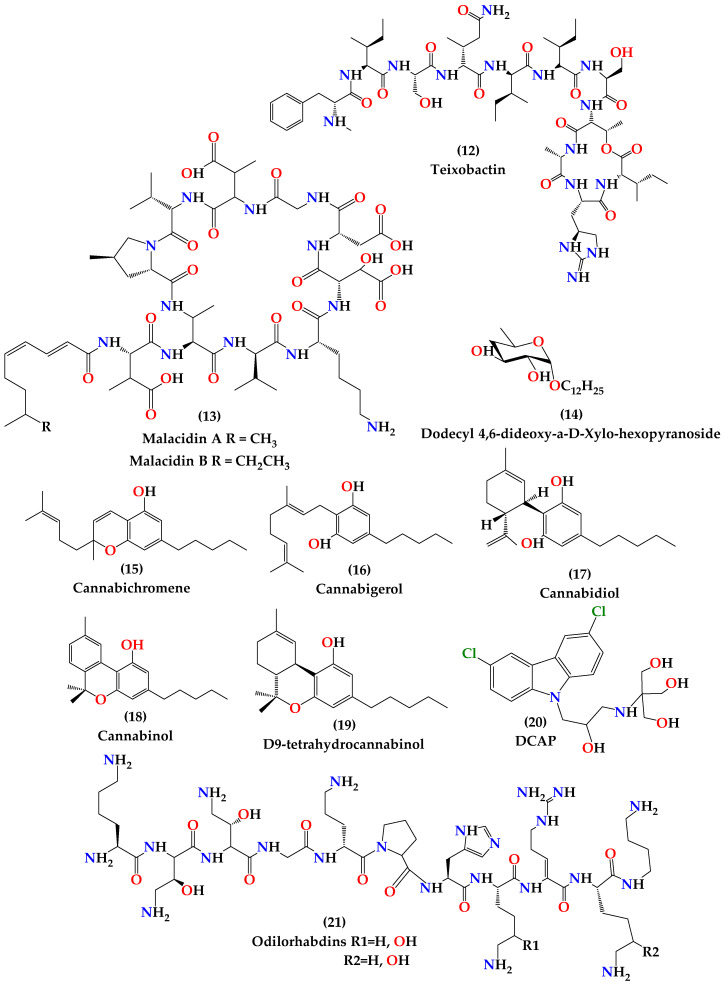
Chemical structures of teixobactin (12), malacidins (13), dodecyl 4,6-dideoxy-α-D-Xylo-hexopyranoside (14), cannabichromene (15), cannabigerol (16), cannabidiol (17), cannabinol (18), ∆9-tetrahydrocannabinol (19), DCAP (20), and Odilorhabdins (21).

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
