# Peer review of "Resistance of Gram-Positive Bacteria to Current Antibacterial Agents and Overcoming Approaches"

_molecules, 2020, doi:10.3390/molecules25122888_

Round 1

Reviewer 1 Report

The manuscript reviewed several species of gram-positive bacteria, the mechanisms of antibiotic resistance of these bacteria, and recent development of treatments based on literature. The manuscript provides some useful information about antibiotic resistance of gram-positive bacteria. However, the review manuscript at its' current stage can be improved before considering for publication with the journal.

Major comments:

  • The review focused on Staphylococcus, Enterococcus, and Streptococcus according to WHO's list. Because the title of manuscript states G+ bacteria, would be possible to review any other G+ bacteria after the three major species?
  • The authors discussed Staphylococcus' resistance to methicillin, vancomycin and other antibiotics. For Enterococcus and Streptococcus, only one drug was discussed. It would be interesting if the review for the two bacteria can be expanded.
  • The title of the manuscript stated "...overcoming strategies" but the text just focused on discussion on some drugs recently reported. It is recommended to conduct a more comprehensive review on what strategies are available to curb the antibiotic resistant G+ bacteria. 
  • More overcoming strategies could be explored by review of different categories of alternatives to antibiotics (e.g. prebiotics, probiotics...., etc.) and their effects and mechanisms; regulatory pathways; prescription and drug use practices; approaches that health care providers, patients, and public can implement to minimize the resistance.

Minor comments:

  • If necessary to have all the figures, they could be better used by adding more related statements in the text and/or in the citations of the figures.
  • Check the journal's guidelines and follow the guidelines when formating text. Example are spaces between brackets and text need to be consistent throughout the manuscript.
  • Proof grammar through the text, examples are spaces needed between words.  

Author Response

Reviewer 1

Major comments

  • Comment: The review focused on Staphylococcus, Enterococcus, and Streptococcus according to WHO's list. Because the title of manuscript states G+ bacteria, would be possible to review any other G+ bacteria after the three major species?

Response: A resistance of additional species of Gram-positive bacteriais was added to the review article under the title of “Other resistant Gram-positive bacteria” (line 240-292).

  • Comment: The authors discussed Staphylococcus' resistance to methicillin, vancomycin and other antibiotics. For Enterococcus and Streptococcus, only one drug was discussed. It would be interesting if the review for the two bacteria can be expanded.

Response:  A review on the resistance of Enterococcus faecium and Streptococcus pneumoniae was expanded to include resistance of Enterococcus faecium toward ampicillin/penicillin and cephalosporins (line 185-191) and other antibiotics (line 206-217), and resistance of Streptococcus pneumoniae to other antibiotics (line 231-237)

  • Comment: The title of the manuscript stated "...overcoming strategies" but the text just focused on discussion on some drugs recently reported. It is recommended to conduct a more comprehensive review on what strategies are available to curb the antibiotic resistant G+ bacteria. More overcoming strategies could be explored by review of different categories of alternatives to antibiotics (e.g. prebiotics, probiotics...., etc.) and their effects and mechanisms; regulatory pathways; prescription and drug use practices; approaches that health care providers, patients, and public can implement to minimize the resistance.

Response: More overcoming approaches and strategies were discussed under the following titles: Bacteriophage therapy (line 506-529), The probiotic approach to prevent antibiotic resistance (line 530-559), National and international approaches and commitment (line 560-579), and Education of prudent antibiotic use (line 580-592).

Minor comments

  • Comment: If necessary to have all the figures, they could be better used by adding more related statements in the text and/or in the citations of the figures

Response:  Textual references for Figures 1 and 2 were modified (lines 44 and 61)

  • Comment: Check the journal's guidelines and follow the guidelines when formating text. Example are spaces between brackets and text need to be consistent throughout the manuscript.

Response: The text’s formatting was checked and revised.

  • Comment: Proof grammar through the text, examples are spaces needed between words.

Response: Missing spaces were added throughout the text.

Reviewer 2 Report

Report for molecules #-827546

Titled: Resistance of Gram-positive bacteria to current antibacterial agents and overcoming approaches

In general, the topic is very important and has a great impact on the medical field. I found the paper to be overall well prepared. My great concern is the lack of some citations. I think the authors need to add the corresponding citations wherever it is necessary, and this comment should be applied throughout the manuscript.

Detailed comments:

Abstract: Authors need to avoid using personal pronoun (eg. We, our ..etc)

Line 21: - please change we provide a review of the most critical … to this article aimed to provide a review of the most critical

Introduction:

Line 53 add a space after 1. To be Figure 1. A diagram

Line 56: Please change the sentence Gram-positive mechanism of resistance to… The resistance mechanism of Gram-positive bacteria (Figure 2)

Line 62-82 some statements need to be supported by reference: -

Line 67 please add a citation

Line 70 add citation

Line 73 add citation

Line 77 add citation

Line 80 add citation

Line 82 add citation

Line 85 add a space after 2. To be Figure 2. A diagram

Also

Line 122 add a citation

Line 125 add a citation

Line 129 add a citation

Line 151 add a citation

159 add a space before It

162 add a space between 30 and S…30 S

Line 165 add a citation

Line 174 add a citation

236-237 add the # after each chemical compound

Line 250: add the # after each chemical compound

270 add a space after malacidins

415-417 add the # after each chemical compound

Author Response

Reviewer 2

  • Comment: Authors need to avoid using personal pronoun (eg. We, our ..etc)

Line 21: - please change we provide a review of the most critical … to this article aimed to provide a review of the most critical

Response: Sentences containing personal pronouns were corrected as advised (Lines 21 and 338)

  • Comment: Line 53 add a space after 1. To be Figure 1. A diagram

Response: Revised (line 55)

  • Comment: Line 56: Please change the sentence Gram-positive mechanism of resistance to… The resistance mechanism of Gram-positive bacteria (Figure 2)

Response: Revised (line 58)

  • Comment: Line 62-82 some statements need to be supported by reference: -

Response: References are added in lines: 69, 71, 73, 77 and 80.

  • Comment: Line 67 please add a citation

Response: Citation is added (line 69).

  • Comment: Lines 70, 73, 77, 80, 82 add citation

Response: Citation is added in lines 71, 73, 77, 80, and 82.

  • Comment: Line 85 add a space after 2. To be Figure 2. A diagram

Response: Done (line 84)

  • Comment: Lines 122, 125, 129, 151 add a citation

Response: Citation is added (lines 129 and 151)

  • Comment: 159 add a space before It

Response: Done

  • Comment: 162 add a space between 30 and S…30 S

Response: Done

  • Comment: Lines 165 and 174 add a citation

Response: Citation is added (lines 165, 169, 173 and 176)

  • Comment: 236-237 add the # after each chemical compound

Response:  The number of each compound was added after the name of the compound (line 319-320)

  • Comment: Line 250: add the # after each chemical compound

Response: The number of each compound was added after the name of the compound (line 331)

  • Comment: 270 add a space after malacidins

Response: Done

  • Comment: 415-417 add the # after each chemical compound

Response: The number of each compound was added after the name of the compound (line 502-504)

Round 2

Reviewer 1 Report

The authors addressed my previous comments adequately. The addition of antibiotic resistance of more gram-positive bacteria, more potential antibiotic alternatives, and strategies to combat antibiotic resistance and relevant references brought more value to the manuscript. With these said, I have only few minor comments:

*In the section of 4.4., because the authors discussed the importance of surveillance of antibiotic resistance, the manuscript will bring broader information to readers if the authors can mention the National Antimicrobial Resistance Monitoring System (NARMS) of the U.S. This is one of the most comprehensive surveillance system in the world that survey antibiotic resistance of bacteria from retail food, humans, and animals.

*It is necessary to proof the grammar and spelling carefully throughout the text. Just an example, the first letter of 'one' under section 4.5. needs to capitalized. Another example, a space is needed between 'approaches' and 'and'in the subheader line of 4.4.

*Check the guidelines of the journal, it needs to consistent throughout the text whether or not spacing between parentheses and letters.

Author Response

Molecules-827546-1- reviewers' comments’ responses

Reviewer 1- Revised 2

  • Comment: In the section of 4.4., because the authors discussed the importance of surveillance of antibiotic resistance, the manuscript will bring broader information to readers if the authors can mention the National Antimicrobial Resistance Monitoring System (NARMS) of the U.S. This is one of the most comprehensive surveillance system in the world that survey antibiotic resistance of bacteria from retail food, humans, and animals.

Response: The National Antimicrobial Resistance Monitoring System (NARMS) of the U.S is now mentioned in section 4.4 as an example of the national approaches to combat bacterial resistance (line 575-578).

  • Comment: It is necessary to proof the grammar and spelling carefully throughout the text. Just an example, the first letter of 'one' under section 4.5. needs to capitalized. Another example, a space is needed between 'approaches' and 'and' in the sub-header line of 4.4.

Response: Spelling and grammar is proofed. Probably, these errors are a result of opening the Word file in different Microsoft Office versions.

  • Comment: Check the guidelines of the journal, it needs to consistent throughout the text whether or not spacing between parentheses and letters.

Response:  The text was checked to be consistent with the guidelines of the journal.
